# A Comprehensive Updated Review on Magnetic Nanoparticles in Diagnostics

**DOI:** 10.3390/nano11123432

**Published:** 2021-12-17

**Authors:** Pedro Farinha, João M. P. Coelho, Catarina Pinto Reis, Maria Manuela Gaspar

**Affiliations:** 1Research Institute for Medicines, iMed.ULisboa, Faculty of Pharmacy, Universidade de Lisboa, Av. Prof. Gama Pinto, 1649-003 Lisboa, Portugal; pedro-farinha@campus.ul.pt; 2Instituto de Biofísica e Engenharia Biomédica (IBEB), Faculdade de Ciências, Universidade de Lisboa, Campo Grande, 1749-016 Lisboa, Portugal

**Keywords:** magnetic nanoparticles, magnetic resonance imaging, magnetic separation, iron oxide nanoparticles, multimodal imaging

## Abstract

Magnetic nanoparticles (MNPs) have been studied for diagnostic purposes for decades. Their high surface-to-volume ratio, dispersibility, ability to interact with various molecules and superparamagnetic properties are at the core of what makes MNPs so promising. They have been applied in a multitude of areas in medicine, particularly Magnetic Resonance Imaging (MRI). Iron oxide nanoparticles (IONPs) are the most well-accepted based on their excellent superparamagnetic properties and low toxicity. Nevertheless, IONPs are facing many challenges that make their entry into the market difficult. To overcome these challenges, research has focused on developing MNPs with better safety profiles and enhanced magnetic properties. One particularly important strategy includes doping MNPs (particularly IONPs) with other metallic elements, such as cobalt (Co) and manganese (Mn), to reduce the iron (Fe) content released into the body resulting in the creation of multimodal nanoparticles with unique properties. Another approach includes the development of MNPs using other metals besides Fe, that possess great magnetic or other imaging properties. The future of this field seems to be the production of MNPs which can be used as multipurpose platforms that can combine different uses of MRI or different imaging techniques to design more effective and complete diagnostic tests.

## 1. Introduction

Diagnosing diseases is the first step toward an adequate treatment. It is essential to understand a patient’s medical and family history, risk factors, their symptoms (or the lack thereof) and cross-check it with the information provided by diagnostic tests in order to correctly deduce their current condition and how it may progress in the future. However, the stage at which a disease is diagnosed also plays a major role in patient prognosis. One of the largest contributors to “avoidable deaths” is the fact that many critical pathologies are diagnosed at too advanced stages. Cancer is possibly the most popular example. A published study from 2009 concluded that a large number of avoidable cancer deaths were due to late diagnosis and consequent delay in potentially curative treatments [1]. Similarly, in the same year, Virnig et al. published a study analysing the disparities in cancer survival between the African American and White populations in the United States of America (USA). Overall, African Americans were more likely to be diagnosed with cancer at latter stages and, simultaneously, less likely to survive longer than 5 years after diagnosis [2]. The same reasoning applies to the treatment of infectious diseases; the determination of the pathogen causing the infection allows the selection of the most appropriate therapeutic options with less potential for antibiotic resistance [3,4]. The examples go on and extend throughout all areas of medicine and nowadays an early diagnosis has become increasingly synonymous with a good prognosis [5]. One important cause for late diagnosis originates from the fact that accurate diagnostic assays are still lacking. For instance, Magnetic Resonance Imaging (MRI) is a commonly used diagnostic test with high resolution and deep tissue penetration. Unfortunately, it has low sensitivity and specificity [6]. On the other hand, Polymerase Chain Reaction (PCR) tests are a highly sensitive and specific assay but it takes too long to obtain a result [7].

Magnetic nanoparticles (MNPs) emerge as potential solutions to the aforementioned problems. Regarding the area of molecular diagnosis, they are presented as promising tools that can be utilized to develop faster, simpler and cheaper diagnostic tests through the application of magnetic separation processes. MNPs can also be used to improve the sensitivity and specificity of MRI. In fact, they have been studied as diagnostic agents for decades with some MRI contrast agent formulations receiving regulatory approval from as early as 1993 [8].

The goal of this review is to present an overview of the main characteristics of MNPs and their applications in medicine, particularly magnetic resonance imaging and magnetic separation; give insight on the current state of MNP research and existing formulations on the market, discuss the technological advances used to improve upon their limitations and comment on how these particles could be applied in the future as diagnostic tools. Figure 1 represents the main objectives of the present review.

This review is separated into different sections. Firstly, the research and information gathering methodologies adopted are presented, as well as the criteria used to determine which publications contained the information with the most interest. The following section focuses on the main characteristics of MNPs as well as their applications in the area of diagnostics, with a focus on imaging and molecular diagnosis. Subsequently, the various inorganic nanoparticles currently being investigated are addressed in more detail, starting with IONPs. These are by far the most studied MNPs to date and therefore an entire section was dedicated to exploring their properties, synthesis methods, existing formulations on the market or under clinical trials and limitations. Furthermore, other inorganic MNPs and their potential to improve upon the limitations of IONPs were examined. Lastly, the conclusions were provided.

## 2. Materials and Methods

The research to carry out this literature review was based on electronic resources. The Pubmed database was the main source of information, complemented by other information sources such as Research Gate and Google Scholar, as well as the use of official sources from the European Medicines Agency (EMA) and the United States Food and Drug Administration (FDA). The research started on 9 December 2020 with the aim of conducting a literature review and ended on 1 September 2021. Research was occasionally carried out outside these dates. From the articles collected from the initial literature search, an analysis was carried out to select the most relevant ones.

Research keywords included: “magnetic nanoparticles”, “magnetic nanoparticles/diagnostics”, “magnetic nanoparticles/imaging”, “magnetic nanoparticles/MRI”, “magnetic nanoparticles/magnetic separation”, “magnetic nanoparticles/applications”, “magnetic nanoparticles/iron oxide nanoparticles”, “gadolinium magnetic nanoparticles”, “manganese magnetic nanoparticles”, “cobalt magnetic nanoparticles”, “silica magnetic nanoparticles”, “dysprosium magnetic nanoparticles”, “magnetic nanoparticles/lanthanide”, “copper magnetic nanoparticles”.

The inclusion criteria considered for the selection process were language (Portuguese, English or Spanish), the title, the abstract and the publication dates, encompassing articles from the last 10 years. Earlier publications with information of interest were occasionally also included.

## 3. Magnetic Nanoparticles

### 3.1. Main Properties/Characteristics

From a general point of view and in the medical field, nanoparticles are colloidal systems sized below 1 µm and preferably from 1 to 100 nm [9,10,11,12,13]. They have been widely studied as drug delivery systems for application in preventing and treating diseases [9]. In addition, depending on their constituents, some of these systems present magnetic properties which open a new realm of potential applications for these particles [9,14,15].

The properties of MNPs strongly derive from their physicochemical characteristics, mean size and morphology. For instance, size influences the intensity of the particle’s magnetic properties. MNPs are composed of regions called magnetic domains each presenting a magnetic moment towards a different direction. In this state, the MNPs do not exhibit magnetic properties. However, when an external magnetic field is applied, the domains align with the field and the particles become magnetised gathering near the field. Similarly, when the magnetic field is removed, the particle returns to its non-magnetised state. This ability to go back and forth is called superparamagnetism. Figure 2 illustrates the superparamagnetic properties of MNPs when exposed to an external magnetic field (B_0_). Unlike bulk magnetic materials, which stay magnetised even after the external magnetic field is removed, only nanoparticles can achieve superparamagnetism. This is an important characteristic of MNPs because, since MNPs do not stay magnetised, they will not aggregate and form clusters [16].

Superparamagnetic MNPs can maintain their colloidal stability and dispersibility, which is important for biomedical applications [16]. Usually, MNPs should have a mean size below 100 nm in order to exhibit superparamagnetic properties [17]. In addition, MNPs with low mean size also present pharmacokinetic advantages. Indeed small sized nanoparticles exhibit better diffusion and distribution towards targeted sites and are less likely to be captured by macrophages [17]. Based on the above described much of the focus on MNP research is being performed aiming to design and synthesize MNPs with high magnetic power with appropriated mean size, preserving their superparamagnetic properties. To achieve this, several approaches have been tested, such as, modifying MNP morphology and adding new elements like zinc (Zn) or cobalt (Co) to the particle’s crystalline structure in a process named doping [18,19,20].

Other crucial characteristics of MNPs include their high surface-to-volume ratio and their ability to bind reversibly to various biomolecules quickly and effectively. This allows MNPs to be efficiently functionalised with a great variety of different coatings to improve their stability, biodistribution profile, among others [7,9,17]. MNPs also exhibit what is called the magnetocaloric effect. When placed under an external alternating magnetic field MNPs can be excited to generate heat. This property can be explained by several phenomena. Alternating magnetic fields cause the MNPs to magnetize and demagnetize in two opposite directions and, through Néel relaxation, energy is released every time the magnetic dipole of a particle flips between two stable orientations within a magnetic field. Heat may also be released through Brownian relaxation, which involves random collisions between particles as well as their physical rotation within a magnetic field (frictional heating). In metallic MNPs, which are excellent thermal conductors, this generated heat can be exchanged with the surrounding environment with the goal of dealing irreparable damage to tumour tissues and induce cell death. This property can open up an interesting application for MNPs as therapeutic options for cancer treatment [10,17].

MNPs are usually grouped into three classes, namely single metal MNPs, metal oxide MNPs, alloy MNPs. Single metal MNPs t present a core composed of one single pure metal structure such as iron (Fe), Co and nickel (Ni). Metal oxides essentially include iron oxides (Fe_x_O_y_) and ferrites, such as CoFe_2_O_4_, MgFe_2_O_4_ or MnFe_2_O_4_. Alloy MNPs consist of a combination of two or more different pure metals, for example, iron cobalt alloys (FeCo) and iron platinum alloys (FePt) [7,21]. MNPs are easy to produce at low cost and are generally biocompatible [16,22,23], though they have some catalytic activity [24].

After intravenous (IV) administration, MNPs can be guided through the application of a magnetic field thus allowing their accumulation at a specific target site [23]. Nevertheless, an overdose of released Fe ions may cause some harm upon long-term exposure [20,21,25]. MNPs in the bloodstream may suffer opsonization from plasmatic proteins, capture from macrophages of the Mononuclear Phagocytic System (MPS) and rapid removal from blood circulation. It is therefore not surprising that MNPs tend to accumulate in the liver, spleen and bone marrow, highly vascularized organs with leaky blood vessels where the MPS is particularly active [20,21,22,23,26]. Once degraded by the immune system, the resulting metal components will either be absorbed by the body or excreted. In vivo toxicity of MNPs is usually associated to increasing levels of metals in the body, especially Fe since it is by far the most used [27,28,29]. Fe takes part in a Fenton reaction which produces reactive oxygen species (ROS), highly dangerous to cells. There are specific proteins that store Fe, however, if they become saturated, the higher Fe levels will induce the production of ROS resulting in cell damage [21,23].

Nevertheless, several studies have been conducted highlighting the safety of metallic MNPs. The rationale is to assess if body natural mechanisms are able to eliminate the molecules in a safe manner. In fact, some metal-based nanoparticles have reached clinical trial and have been approved for commercialization [30,31,32]. Nevertheless, some studies have also concluded that MNPs, with different applications, induce different degrees of toxicity. For example, in hyperthermia cancer treatments, repeated administrations of MNPs are required to ensure they remain at affected tissues thus maximizing the therapeutic effect [21]. The prolonged accumulation of MNPs might have a direct impact on higher particle degradation and consequent release of its metal components leading to greater toxicity symptoms [21].

On the opposite side, MNP formulations used as contrast agents only tend to be used on the same patient from time to time thus allowing the body the possibility to eliminate the particles without reaching high levels of metal in the blood [21]. This disadvantage can be overcome by coating the nanoparticle surface with a ligand or antibody. This is one of the most common techniques used to increase their specificity and accumulation at target sites. For instance, coated MNPs have been used to target cancer cells or amyloid plaques in patients with Alzheimer’s disease [17,33,34]. Another reason for functionalization is to reduce MNPs’ clearance via the MPS. To decrease their clearance and, at the same time, improve biocompatibility in vivo, MNPs can be functionalized with polymers such as poly(ethylene glycol) (PEG) which is known for gifting nanoparticles with stealth properties thus escaping their recognition by the MPS [20,23].

Lastly, it has been extensively described the use of MNPs associated with polymer or lipid nanoparticles. These hybrid systems combine the safety of organic molecules with the magnetic properties provided by inorganic substances. This can be done by encapsulating the magnetic molecules in organic nanoparticles, preventing the leakage of metal ions in toxic quantities [35,36,37].

### 3.2. Application of MNPs in Medicine

One of the great advantages of MNPs is the wide range of potential applications. For one, they can be used as therapeutic agents in various manners, such as active drug delivery (cytotoxic, antimicrobials and gene delivery) into specific regions of interest in a controlled manner, such as cancer tissues, guiding them through the body using an external or internal magnetic field [9,16,17,20,38,39,40,41,42]. MNPs can also be used to treat iron deficiency anaemia (IDA) in patients with chronic kidney diseases (CKD) aiming to increase Fe levels in the bloodstream [31,43,44]. Furthermore, MNPs have been studied for bone tissue repair and engineering. The rationale of this approach is to combine the application of an external magnetic field (which has been proven to aid in hard tissue repair) and deliver magnetic nanoparticles with tissue engineering scaffolds or stem cells in order to maximize osteogenic differentiation, biomineralization and tissue regeneration [41,45,46]. Lastly, another therapeutic possibility for MNPs is called “hyperthermia cancer treatment” in which an alternating magnetic field is applied inducing the nanoparticles to spin and generate temperatures above 40 °C at tumour sites, damaging cancer tissues [17,39,40].

Aside from therapeutic options, MNPs are also quite promising tools for disease screening and diagnosis. One of their most popular applications of MNPs comes in the form of contrast agents for MRI. By upgrading the technique’s sensitivity and specificity it is possible to obtain more discernible images that can provide a more reliable diagnosis as early as possible [9,16,20,21,39,40,41,42]. Additionally, they play an important role in biosensors that may be used to detect specific biomarkers of inflammation or cancer, in the magnetic separation of biomolecules for molecular diagnosis purposes, as well as in enzyme immobilization [16,41,42]. This review is focused on the recent advances of MNPs use in disease diagnosis, particularly in the areas of imaging (MRI) and magnetic separation of biomolecules. In Figure 3 are depicted some biomedical applications of MNPs.

#### 3.2.1. Imaging

MRI is a non-invasive diagnostic assay that allows detailed images of a patient’s soft tissues and is widely used in medicine today [30,47]. When a magnetic field (designated as B_0_) is applied the hydrogen protons in the body, which were previously randomly oriented, will align accordingly. Most protons, also designated as spins, will point in the direction of the magnetic field (low energy spins) whilst a smaller number of energetic protons will point against the magnetic field (high energy spins). However, MRI exams capture the magnetic behaviour of millions of protons, and therefore, instead of looking at each spin individually it is more appropriate to look at the averaged sum of groups of protons. This sum is called the net magnetisation (M) and it can be treated as a regular vector of classical physics. When B_0_ is applied the vector M points in the direction of the magnetic field reaching an equilibrium and is called equilibrium magnetization M_0_ [47,48,49]. Then, a radiofrequency pulse is applied disrupting M_0_, forcing it to spin out of equilibrium. Once the radiofrequency pulse is stopped M returns to the equilibrium state through what is called the relaxation phenomenon [40,47]. The time that M takes to return to the M_0_ state, as well as the electromagnetic energy released from the relaxation process, is then measured and a computer converts this information into detailed images that can be visualized by the medical examiners [47]. The process is schematically represented in Figure 4.

There are two relaxation processes: the longitudinal or spin-lattice relaxation (T1—recovery) and the transverse or spin-spin relaxation (T2—decay). T1 relaxation involves the energy exchange between the spins and their surrounding environment producing images with a bright signal. Some regions of the body generate brighter signals compared to their surroundings and thus can be more easily identified, this is known as positive contrast. On the other hand, T2 relaxation is a result of energy transferring between spins which results in loss of phase coherence and produces images with a dark signal. Unlike positive contrast, when certain regions produce darker signals compared to their surroundings, they can be visualized through negative contrast. Another parameter is the apparent transverse relaxation time, or T2*, in which the T2 phase loss is enhanced due to inhomogeneities in the applied magnetic field making T2* relaxation time shorter than T2 [16,47].

Different tissues will have varying relaxation times and, therefore, will produce different images. These differences stem from the tissues’ proton density and physicochemical properties [47,50]. For instance, water molecules are small, rotate rapidly and the electronegative oxygen atom pulls the electron clouds away from the hydrogen nuclei, exposing them to the external magnetic field. On the other hand, fat molecules tend to be bulky, rotate slowly and have their hydrogen nuclei nestled within long carbon chains, shielded from the external magnetic field by electron clouds. As a result, the relaxation phenomenon in each tissue will be distinct. For example, adipose tissues present shorter T1 relaxation times compared to tissues with a higher water content [51,52]. These differences among tissues allow a distinction between them, and more importantly, it enables visualization of potential tumours, strokes and inflammation sites [20,53] thus helping to diagnose the extent of certain pathological conditions [47,54,55,56]. For example, MRI can be used to distinguish between ischemic or haemorrhagic strokes and understand the full extent of the damaged area [57,58], localize metastasis and understand the size and stage of the tumour [9,55,59,60], diagnose demyelinating diseases [56,61,62], Alzheimer’s dementia [34,63,64,65], congenital heart diseases [66,67,68,69], among others [47,70,71,72].

Since it does not involve ionizing radiation, MRI possesses a safety advantage over Computed Tomography (CT) and Positron Emission Tomography (PET) scans, whilst also presenting a higher spatial resolution compared to PET. Sometimes these imaging techniques can also be combined and applied simultaneously aiming to provide more complete information about the patients’ health status. The most common is the PET/CT scan, which provides an accurate diagnosis at a lower cost than PET/MRI scans [73]. However, PET/MRI has been proven to possess various advantages compared to PET/CT: (1) higher soft tissue resolution; (2) lower radiation exposure that can even reach an 80% decrease, thus constituting a safer option to prevent the appearance of neoplasia, especially in the paediatric population [73].

Although further research is necessary PET/MRI exhibits superior diagnostic capabilities [73,74]. In addition, MRI is available in many medical institutions, therefore easy to apply [75]. Although no health risks associated with the magnetic field or the radio waves applied have been described, each patient must be thoroughly screened for the presence of implants and devices prior to MRI examination to prevent unnecessary injuries [56,75]. However, there are patients who present adverse reactions to some contrast agents. Even though MRI is capable to generate high-contrast images of soft tissues, sometimes it is necessary to associate contrast agents. Contrast-enhanced MRI is able to further highlight the anatomic and pathologic features of regions of interest and consequently achieve better results. The most commonly used contrast agents are based on gadolinium (Gd), a rare metal with excellent T1-weighted imaging properties [30,75,76]. Nevertheless, patients with severe renal failure requiring dialysis may risk nephrogenic systemic fibrosis (NSF) upon receiving Gd-containing agents. Therefore, the risks and benefits should be evaluated case by case [30,56]. There are also patients who develop anaphylactoid reactions to contrast agents, especially Gd-based contrasts, but these cases are few, and the benefit-risk ratio is still positive. Moreover, these side effects might occur in patients who have also demonstrated allergic reactions to other contrast agents, such as iodine-based contrasts, and so a thorough medical background check can reduce a significant percentage of the risk [75].

At present, the major drawbacks involving MRI tend to be related to its low specificity and increased cost when compared to other imaging techniques such as CT. Breast cancer screening is a perfect example to show the downfalls regarding MRI. Although its use has increased over the years, the cost of MRI limits its use to high-risk groups and lesions that are difficult to detect using standard imaging techniques. Instead, medical professionals choose screening tests that are cheaper, quicker, and easier to perform, such as ultrasound and mammography [9,60]. Furthermore, its low specificity sometimes results in the identification of false positives and consequently the application of unnecessary chemotherapeutic treatments [9,60].

In this regard, MNPs constitute a promising tool in combating these weaknesses by improving both sensitivity and specificity of MR imaging techniques. MNPs are able to accumulate at the desired tissues and shortening the proton’s relaxation time thus enhancing the MR image. T2 relaxation times are very dependent on the existence of MNPs, since most of them are Fe_x_O_y_ based and most IONPs mainly influence T2 relaxation. This occurs because MNPs possess their own magnetic fields thus creating inhomogeneities in the overall applied magnetic field which shortens T2 relaxation times. Shorter relaxation times produce more detailed darker images thus increasing the sensitivity of the method. Taking into account this evident T2 effect, MNPs have been more commonly used in negative contrast enhancement using T2-weighed pulse sequences [16,39,40,47,53,77,78].

However, many MNP-based contrast agents have also proven to be effective in performing T1-weighed MRI, for example, nanoparticles containing Gd or manganese (Mn), [15,33,79] and extensive investigation has also led to the development of MNPs that significantly shorten both T1 and T2 relaxation times [15,80]. The latter is referred to as dual mode T1- and T2-weighted MRI and has the potential to increase the diagnostic accuracy of MRI. Unlike other multimodal imaging techniques (e.g., PET-MRI or MRI-CT), which will be touched upon afterwards, this dual mode imaging generates matching T1- and T2-weighted images using a single instrumental system. The information provided by each image complements the other thus allowing radiologists and other imaging physicians to get a more complete picture of the patient’s status and achieve a more accurate diagnosis. Dual mode contrast agents can be produced through strategies including metal doping of IONPs or by synthesizing MNPs with a T2-weighted contrast core and a T1-weighted contrast shell. These MNPs could potentially provide two simultaneous and complementary images and improve the diagnostic accuracy of MRI [15,81,82].

Besides simple MRI contrast enhancement, MNPs can also be applied as multimodal imaging platforms. For example, the method clinically used for PET-MRI imaging involves sequential injections of PET and MRI agents [14]. However, each contrast agent has its own pharmacokinetic attributes which lead to unmatched PET and MRI images. To overcome these limitations, studies have been conducted using MNPs [83] coated with radioisotopes used for PET imaging. The objective is to take advantage of PET’s high sensitivity and MRI’s high resolution and combine both techniques into a single more complete examination with overlapping images. Many examples of MNP formulations have completed clinical trials and were approved for clinical use as contrast agents for MRI [34].

#### 3.2.2. Molecular Diagnosis

Molecular detection systems are one of the most common applications of MNPs when it comes to diagnosis. Their surface chemistry and magnetic properties play a large role in developing quicker and more effective methods for analytical procedures.

This section will focus on the specific applications of MNPs in the field of molecular diagnosis, including: (1) DNA and RNA detection and separation; (2) protein purification; and (3) cell separation [7,84,85].

##### Nucleic Acid Separation and Detection

Nucleic acids are one of the most important biomolecules in the human body and their functions mainly include storing, copying and transmitting genetic information. As such, when changes and mutations occur it may predispose and lead the population to the development of various diseases, including type 1 (T1D) [86] and type 2 diabetes mellitus (T2D) [87], Alzheimer’s dementia [88,89,90], cystic fibrosis [91] and various types of cancer [92,93,94].

Detection and identification of these genetic changes are important steps in predicting the risk of developing a health condition, choosing an appropriate treatment and evaluating a patient’s prognosis [95,96,97]. The same rationale can be applied for the identification of the genetic information of various pathogens enabling the diagnosis of infectious diseases [7,85,98].

To this date, the gold standard used in the detection of nucleic acid is PCR, which even though it is highly sensitive and specific, it still has significant limitations that hinder its use [99]. Firstly, it is both time-consuming and laborious, requiring hours to get a result [84,100]. The extraction process is frequently accompanied by long drawn-out consecutive centrifugation steps which typically result in low yields with suboptimal purity [7]. Secondly, PCR requires large and expensive equipment [100] which can only be operated by professionals with adequate training [99,100,101]. This results in higher logistical costs regarding PCR consumables, machinery and the facilities limiting the use of PCR to institutions that possess more funding and better equipped facilities [7].

Therefore, it has become necessary to develop molecular nucleic acid detection procedures, which while maintaining accurate outcomes, will produce results in a short period of time and that do not require expensive equipment or highly trained personnel. The inherent properties of MNPs, such as, high surface-to-volume ratio, magnetically controlled particle aggregation and dispersion [102] and their ability to bind to a vast number of different biomolecules (DNA, RNA, enzymes) offer potential solutions that can be used to isolate DNA and RNA from complex samples as well as enriching nucleic acid concentration, a useful tool that facilitates their detection [7].

The extraction process is the first step in molecular analysis of nucleic acids and cannot be overstated. It is essential to the overall detection and analysis process as it has a direct influence on the downstream steps [7]. In this process, MNPs bind to the nucleic acids, achieved through functionalization of said MNPs with ligands that specifically bind to DNA and RNA, and are then separated from the remaining sample matrix by applying a magnetic field through what is known as the magnetophoretic phenomenon. During this step, the MNPs, which are still bound to their target, are gathered towards the magnet making it easy to discard the unwanted material. The MNPs are then washed in an elution buffer to promote the release of the DNA/RNA molecules from the nanoparticles. Afterwards, the MNPs are separated from the supernatant containing the free nucleic acid molecules using an external magnetic field [7,84,103]. Figure 5 presents a schematic overview of the magnetophoretic phenomenon [103].

One of the greatest advantages of using magnetic beads is the fact that their aggregation does not require centrifugation nor column separation steps and, therefore, shortens the time it takes to separate the components as well as removing the need for costly equipment, such as centrifuges and liquid chromatography systems [102,103,104].

Furthermore, magnetic separation allows for recycling of the magnetic beads and is very well suited for large-scale use. All of these factors combined is what enables a quicker, more efficient and cheaper method for nucleic acid separation and purification [7,16,102]. On the other hand, one problem encountered in magnetic separation is the possible low release of the captured nucleic acids from de MNPs. Generally speaking, in order to optimize the absorption ratio, MNPs tend to be coated with positive charges (usually amino groups) since they easily form ionic bonds with the negatively charged phosphate groups present in nucleic acids. However, these strong bonds can also hinder the nucleic acid desorption process. Nevertheless, several strategies have been implemented in order to minimize this effect. Studies by Rutnakornpituk et al. and Kang et al. concluded that desorption efficiency rises with higher concentration of elution buffers and less positively charged MNPs [105,106]. Meanwhile, Tanaka et al. conducted similar studies and concluded that the choice of buffer also influences the dissociation process [107]. Other studies also examined different parameters, including temperature, application of external magnetic fields, mesoporous MNPs and organic solvents. By optimizing these factors it is possible to design MNP-based nucleic acid separation protocols for maximum efficiency [108,109].

##### Protein Purification

The ability to isolate, purify and actively manipulate proteins and peptides has become of great importance in the field of biotechnology [16]. Traditional protocols often involve electrophoresis, ultrafiltration, precipitation and chromatography, with the latter standing as the choice of election when it comes to efficiency and selectivity [16,110,111]. However, chromatography is time-consuming and mostly restricted for use in pre-treated solutions given that inhomogeneous protein mixtures are incompatible with the particulate-free conditions required [112,113].

Magnetic separation is considered as a potential alternative to isolate proteins from complex samples. By applying the magnetophoretic phenomenon previously explained in section “Nucleic Acid Separation and Detection” MNPs can be used to separate and purify proteins and peptides through methods that are straightforward, cheap, fast and easily scalable without the need to employ dedicated equipment such as centrifuges, filters and liquid chromatography systems [16]. It also stands as a non-destructive separation method, which contributes to maintaining the proteins’ structural integrity [110]. Furthermore, compared to chromatography, magnetic separation stands as a faster, more versatile method that can be employed in samples without pre-treatment, is cost-effective and allows for reusability of sorbents [42].

##### Cell Separation

Microbial infections have been for a long time one of the most concerning public health challenges worldwide. Even though nowadays infections are a more pressing matter in developing countries, they are still a significant cause of death all over the globe [3,4,38,84,114]. In the food industry, preventing the contamination of food is of the utmost importance. Thus, effective methods are key in detecting and identifying possible contaminant microorganisms [115]. In medical facilities, such as hospitals, bacterial infections have always been a considerable cause of death as well as a major contributing factor to longer patient hospitalizations and increased healthcare costs [3,4,116]. Furthermore, even though antibiotics still remain effective against most bacteria, the cases of antibiotic-resistant bacteria are constantly increasing which results in more deaths, higher costs and an overload on medical professionals [3,4]. Today, one of the best strategies available to deal with this problem is the rational use of antibiotics. However, to do so, physicians must know which bacteria is causing the infection and traditional detection and identification methods take too long to produce a conclusive result [98,115,116].

At present, microbial cultures are still the most used method in bacterial identification and susceptibility tests which raises problems since it can take days to get the results. This method also requires extensive manual labour. It is prone to sampling and enumeration errors in low pathogenic concentrations, such is the case in food samples, and results in delayed diagnosis [98,115,116]. Therefore, there is a need for quicker, yet effective, identification methods in order to initiate proper treatment based on rational antibiotic management in a timely manner [116].

Magnetic separation using MNPs has been studied as a potential tool for improving upon the drawbacks of the currently used techniques [16]. The principle of using MNPs in targeted pathogens is similar to what has been already explored in the previous sections. For this, a magnetic separation usually involves MNPs coated with antibodies or peptides (immunomagnetic separation) that specifically target the pathogens, which are then separated from the remaining sample, a culture medium or a food matrix, through a magnetic gradient [98,115,117]. After separation and concentration, the microorganisms can then be identified through conventional methods, such as PCR, colourimetric, fluorescent and surface-enhanced Raman detections [16,98]. MNPs simplify the pre-enrichment step by aggregating and concentrating the target bacteria into smaller volumes reducing the overall testing time [115], and by isolating the bacteria-bound MNPs from the remaining non-magnetic complex sample environment this method is able to decrease the background noise [16,114,115].

However, the major drawbacks of immunomagnetic separation revolve around the affinity of the antibodies towards the target. On the one hand, there is a risk of antibody cross-reactions which can generate false positive results or increase background signals [115]. On the other hand, antibodies are specific to one or few bacteria strains which complicates any process with the goal of discriminating different bacteria strains simultaneously. Further investigation regarding magnetic immunoassays in complex matrices is required and different strategies need to be tested in order to overcome the current drawbacks [98].

## 4. Iron Oxide MNPs

### 4.1. Properties of IONPs

Amongst the plethora of different MNPs, Fe_x_O_y_ stands as one of the most commonly used materials for the synthesis of MNPs [110,118,119,120]. Iron oxide MNPs (IONPs) are by far the most studied MNPs [118,119,120]. They have attracted a lot of attention as a result of their low toxicity and biocompatibility, high surface-to-volume ratio and superparamagnetic properties. Iron is very well tolerated by the human body, making IONPs possibly the safest option compared to MNPs based on other elements. The three most common forms of Fe_x_O_y_ are magnetite (Fe_3_O_4_), maghemite (γ-Fe_2_O_3_) and hematite (α-Fe_2_O_3_) [9,26,98,121]. These nanoparticles are essentially composed of two parts: (1) an inorganic Fe_x_O_y_ core responsible for the particle’s ferromagnetic properties and (2) external layers of coating, as can be observed in Figure 6.

Since Fe and Fe_x_O_y_ tend to easily suffer chemical oxidation, the outer coating serves to protect the core from degradation, maintaining its integrity and preventing the release of metabolic biproducts derived from the degradation of the MNP [22,26,31,118,123].

Not only do IONPs suffer chemical degradation, but also uptake from the MPS resulting in cellular degradation of the MNP. Coating the nanoparticles with molecules that are more biocompatible and less recognized by the immune system is one of the most common strategies used to improve the half-life of IONPs [22,26,31,118,123].

Polymeric coatings such as PEG, poly(lactic-co-glycolic acid) (PLGA), poly(vinyl alcohol), PLGA-PEG and various other copolymers, alginate, chitosan or dextran are quite common with the general goal of improving biocompatibility and blood circulation times [31,104,110,124]. Dextran, in particular, has been used in many formulations approved for clinical use [16,31]. However inorganic coating, such is the case of silica (which allows functionalization) [26,104,110,124], graphene (granting better thermal stability and electrical conductivity and larger surface-to-volume ratio to further the potential array of applications) [110] and gold (allowing functionalization, biocompatibility and protecting MNPs from oxidation) [26,104,110] have also been used. The MNP coating also prevents particle aggregation and precipitation, as well as granting hydrophilicity, which improves the overall formulation pharmacokinetics [26,110]. Coating can also be used to improve the efficacy of IONPs. Antibodies, targeting ligands, fluorescent dyes, radioisotope tracers, among others, can be functionalized onto the surface of IONPs to increase their target specificity or imaging capabilities [9,22]. Similarly, some coatings are indispensable in the magnetic separation process. For instance, silica and gold can specifically bind to nucleic acids and certain metal complexes selectively bind to certain kinds of proteins, which enables their separation and extraction. Figure 7 provides a schematic representation of some of the different coatings used in MNPs as well as their specific functions.

### 4.2. Synthesis of IONPs

The properties of IONPs are highly dependent on their size, shape and spatial distribution of crystals within the particles. So, to achieve the desired specifications a selection of the most appropriate synthesis method should be performed [22,31].

There are two main approaches available when producing MNPs: top-down or bottom-up. In top-down approaches, the starting metal bulk material/thin film is broken down to the nanometre level creating the nanoparticles, whereas in the bottom-up approach the base Fe_x_O_y_ molecules precipitate and suffer nucleation and growth steps until a nanoparticle is obtained [110]. IONPs can be synthesized through chemical, physical and biological techniques, with the first one being by far the most adopted mainly due to its low production cost, high yields, and the ability to functionalize the nanoparticles [26,125,126]. The most common chemical methods include co-precipitation, thermal decomposition, hydrothermal synthesis, microemulsion, sol-gel synthesis, sonochemical synthesis, electrochemical synthesis, among others and some popular physical methods include spray or laser-induced pyrolysis, laser ablation, milling and lithography [21,26,31,127].

Two previous studies presented an interesting overview of the different methods as well as the frequency with which they are applied (Figure 8) [26,128]. In the Appendix A, two tables are provided for further characterisation of each method. Appendix A offers a general description of the most used synthesis techniques [17,21,26,27,31,45,104,110,124,125,126,127,129,130,131,132,133,134,135,136,137,138,139,140,141,142,143], and Appendix A summarizes the advantages and disadvantages of each method [6,16,17,21,26,27,31,45,104,110,124,125,126,127,130,134,135,136,139,141,144,145,146]. Another important aspect is that synthesis methods are also applied to other MNPs besides IONPs [45,129,130,140,141,142,145,147]. Co-precipitation stands as the most widely utilized method for synthesizing IONPs [26,124,127,135]. In fact, most commercialized IONPs are synthesized via this method [31].

### 4.3. Clinically Approved IONPs

IONPs have been extensively studied as diagnostic tools to detect cancer, heart disease and other inflammation-related diseases, in part by heavily relying on their MR imaging properties [31,146,148].

Many IONP formulations have undergone clinical trials with some having achieved approval for clinical use by both the EMA and the FDA, namely, ferumoxytol, ferumoxide, ferumoxsil, ferristene. Ferucarbotran and Sienna+^®^ are also IONPs approved for clinical use in the European Union (EU), however not by the USA [16,30,31,32,55]. Appendix A presents examples of currently clinically approved IONPs, along with their applications [28,30,31,32,149,150,151,152,153,154,155,156,157,158,159,160,161,162,163].

Ferumoxytol is an interesting case as it is not clinically approved for imaging purposes, but rather for the treatment of IDA in adult patients with CKD. However, it has also been studied, and sometimes used utilized off-label, as an IV contrast agent for imaging of primary tumours, lymph node cancer metastasis, multiples sclerosis, post-transplant renal imaging, and cardiovascular disease in patients with kidney failure at risk of developing NSF [31,32,164]. Although uncommon, NSF is a severely debilitating condition prone to develop in patients with kidney failure who are exposed long term to Gd contrast agents. The safety profile of IONPs suggests they can be used as alternative safer options to Gd compounds in patients with CKD who require MR contrast procedure [31,32,164]. Ferumoxytol is also easy to use as it can be given as a short intravenous bolus for MR angiography (MRA) analysis and dynamic MR, where it works as a blood pool agent [164]. Blood pool contrast agents tend to remain in the intravascular compartment and not easily extravasate into the extracellular space, achieving a high blood half-life. This fact allows imaging of blood vessels to detect abnormalities, such as aneurisms and atherosclerotic plaques and facilitates the measurement of the Blood Brain Barrier’s (BBB) leakiness, which directly correlates to inflammation from brain tumours, trauma or multiple sclerosis [31,164].

Ferumoxide is indicated for IV administration in adult patients as an adjunct to MRI to enhance the T2 weighted images used in the detection and evaluation of lesions of the liver that are associated with an alteration in the MPS [31,32]. This is due to the high hepatic uptake of IONPs by the MPS which allows an accurate visualization of primary lesions or metastases in the liver. In pathological conditions, namely fibrosis, cirrhosis or neoplastic lesions (e.g., hepatocellular carcinoma) the IONP uptake by macrophages is lower compared to healthy tissue [22,31,47]. Furthermore, clinical trials have demonstrated the applicability of ferumoxide in cell tracking (macrophages and mesenchymal stem cells) [32,165,166].

Ferucarbotran is also an IV contrast agent approved for liver imaging, under the names Resovist^®^ and Cliavist^®^ (Germany, Leverkusen). Although another ferucarbotran nanoparticle, Supravist™, has entered phase 3 clinical trials as a positive blood-pool agent [31,32].

Sienna+^®^ is a subcutaneously administered IONP dark magnetic tracer approved for sentinel lymph node (SLN) detection in breast cancer. It is classified as a Class IIa device and is CE-approved for marketing Europe, making it the first marketed nanoparticle device. It works in tandem with another medical device, the Sentimag^®^, a handheld magnetometer used to detect where the tracer is accumulated. By using the magnetometer and visualizing a colour change (SLN colour changes to brown/black with the local accumulation of Sienna+^®^), the SLNs can be accurately identified [157,158]. In a meta-analysis by Teshome et al., Sienna+^®^ was classified as non-inferior to SLN mapping using radioisotope with or without blue dye, the standard technique for detecting SLNs in early-stage clinically node-negative breast cancer [157,158].

There are also two examples of IONPs approved for oral administration, ferumoxsil and ferristene. They are indicated for gastrointestinal and bowel imaging, for the detection of necrosis, oedema, fistulas, tumours, ascites and other abscess formations [31,167]. There is also evidence that oral IONPs can be effectively used as a tool for visualizing the extrahepatic biliary tree, which is required in liver transplants [32,168].

It is also worth stating that various IONPs are currently on the market with applicability in magnetic separation of nucleic acids, proteins/antibodies and cells. Most are composed of solid cores of magnetite and/or maghemite coated with substances that enhance their binding capabilities. These coatings are easily modified according to the target molecule to increase their binding specificity and prevent non-specific binding. Surface layers made of silica for nucleic acid separation; use of specific antibodies for cell immunoseparation; or coating with ionic complexes (copper or zinc ions coupled with nitrilotriacetic acid) for binding to specific proteins are just some of the examples available on the market [169,170,171,172,173,174,175,176,177,178]. In Appendix A are presented various examples of IONPs currently available for commercialization, as well as their specific applications [169,170,171,172,173,174,175,176,177,178].

### 4.4. IONPs in Clinical Trials

There are also other IONPs currently going through clinical trials. Ferumoxtran-10, a contrast agent intravenously (IV) administered, has been used to detect lymph node metastasis (phase III, clinical trials) [159], particularly in prostate cancer. After extravasating into the tissues, the nanoparticles are then cleared by the lymph nodes and eventually taken up by macrophages present within lymph nodes. The presence of metastasis negatively impacts lymph node functions, resulting in less macrophage circulation and, therefore, less ferumoxtran content. The reduced IONP content produce a brighter image, allowing for differentiation between healthy lymph nodes and normal-sized metastatic ones [31,32,158]. However, it has not been approved for the market due to a lack of statistically significant benefit in sensitivity and specificity [179]. Ferumoxtran has also been studied for visualization of the Central Nervous System (CNS) with the goal of detecting brain tumours and other lesions [31,32,164]. The IONPs suffer uptake from macrophages in the CNS and concentrate in areas where inflammation is most prominent, for example, in tumours, ischemic lesions and demyelinating diseases (e.g,. multiple sclerosis). Compared to Gd-based contrast agents, ferumoxtran has demonstrated a prolonged contrast enhancement which could go up to 7 days and showed additional areas in brain tumours that could not be visualized with Gd contrasts [31,158,164].

When studying patients with multiple sclerosis (MS), the use of MRI combining both Gd chelates and IONPs has proven superior to imaging using a single contrast agent and improved the detection of active lesions [31,158,164]. Furthermore, ferumoxtran has been used in imaging of insulitis as a means to early diagnose T1D. In T1D an autoimmune inflammatory response is generated against the pancreatic β-cells, which produce insulin, resulting in lasting damage to the pancreatic islets and a deficiency in insulin production. The pancreas of patients suffering from insulitis presents an enhanced vascular leakiness and a greater concentration of macrophages, caused by the local inflammatory response. This causes the IONPs to extravasate into the inflammation site and be taken up by the macrophages. The accumulation of IONPs leads to enhanced images that help discriminate between healthy and diabetic tissue [31]. Lastly, ferumoxtran has also undergone clinical trials as a blood pool agent for MRA [31,158,164,180].

Another IV administered IONP evaluated in clinical trials for MRA is feruglose. It has been used as a blood pool agent for coronary angiography, in evaluating coronary artery bypass performance and for detection of coronary artery stenosis. It has also shown effectiveness in detecting haemodynamically significant stenoses in iliac, femoral and popliteal arteries [151] and demonstrated high specificity in abdominal and pelvic angiography, although the attendant venous overlap can limit the assessment of stenosis in renal and pelvic arterial segments [152]. Additionally, feruglose was studied for contrast-enhanced venography in patients suffering from deep vein thrombosis. However, feruglose did not show superiority when compared to CT-based radiographic venography [31].

VSOP-C184 (IV) administered is another IONP studied as a blood pool agent. A phase 1 clinical trial deemed VSOP-C184 as a safe, tolerable and effective MRI contrast agent [154] and it was also evaluated for coronary angiography where it achieved a moderate diagnostic accuracy in the detection of coronary stenosis [155].

### 4.5. Limitations of IONPs

Although the potential of IONPs in the medical field is vast, there are still several limitations that hinder their clinical application.

Firstly, they are not very profitable as contrast agents [31,32,44,158]. Most approved IONP formulations have either been withdrawn from the market or are only used in very specific situations. Ferumoxsil, ferristene, ferucarbotran, ferumoxide, are all examples of IONPs approved for commercial use which have later been withdrawn from the market both in the USA and the EU due to lack of users, even though the safety and efficacy was proven [31,32,44,158]. Ferumoxytol has also been withdrawn, however only from the EU market, and is only used in adult patients with CKD requiring treatment of IDA or when Gd-based contrast agents are contraindicated. Gd-based contrast agents tend to be preferred by clinicians over IONPs because they show positive contrast enhancement (T1-enhanced contrast) which is often preferred to the negative contrast enhancement of IONPs because it is easier to visualize a signal enhancement (bright image) than a signal loss (darker image) [158,181]. Additionally, IONP agents must compete with the fast-paced development in MRI technologies and face the slow process of regulatory approval. The latter has particularly negatively impacted the use of Feridex^®^ (ferumoxide) and Resovist^®^ (ferucarbotran) since the delay in clinical approval meant these agents missed the opportunity before the approval of the Gd-based contrast agent Primovist^®^, which became widely used [158].

Unlike most therapeutic agents, which can be administered for days, weeks or for the remainder of a patient’s life, contrast agents are used very scarcely, meaning the financial return for contrast agents tends to be much lower unless they are utilized in bulk. The low IONP sales are not sufficient to cover the costs of pharmaceutical companies and have therefore become almost unavailable to the public, in many cases [31,32,158]. For example, ferumoxide has been taken out of the market since 2008 and ferucarbotran is only available in limited countries, such as Japan [31,32,158]. Not only that, but the clinical development costs of contrast agents are similar to those of therapeutic agents and even if they manage to be approved by the regulatory authorities there is no guarantee that clinical diagnosticians will choose to use them. In fact, most radiologists are not experienced in interpreting IONP enhanced images [158]. Due to this, even though IONPs are available options, most professionals prefer ordering images enhanced with other more common contrast agents, such as Gd [158]. The nanoparticle’s clinical application is also a factor that contributes to less IONPs in development [158]. For example, very low effort is being made by pharmaceutical companies in the area of imaging of stem cell migration and immune cell trafficking, even though the potential of IONPs in these areas is immense. This could probably be due to the low return on investment and the demanding regulatory requirements for the approval of drugs in this particular medical field [158].

Another limitation is directly based on the IONP contrast mechanism. They produce a dark signal and are susceptible to artefacts in MRI, which could arise confusion between hypointense areas or genuine pathological conditions (e.g., early-stage tumours). Moreover, they produce lower contrast when compared to T1-weighted images [110,182]. In practice, this means that IONPs have limited use in regions of the body that naturally produce a low signal, organs with an intrinsically high magnetic susceptibility (e.g., lungs) and in the presence of haemorrhagic events. However, some approaches have been suggested, such as spin-echo sequences, inversion recovery ON-resonant water suppression (IRON)-MRI and the employment of micron-sized IONPs [110]. T2-weighted MRI enhancement is also more susceptible to inaccurate measurement due to magnetic field B_0_ inhomogeneities, although further improvements in MR technology are expected to at least minimize the problem by allowing for a superior quantification accuracy and image resolution. Another approach to counter this problem is to explore T1 relaxation properties of IONPs or the use of dual-modality (simultaneous T1 and T2 relaxation enhancement) contrast agents [158].

## 5. Other Inorganic MNPs

Although IONPs are the most studied, there are many other MNPs currently being developed. As mentioned previously, MNPs can, essentially, be divided into 3 major groups: (1) single metal nanoparticles, (2) metal oxide nanoparticles and (3) metal alloy nanoparticles [47]. However, there are many other examples of magnetic nanosystems that do not belong to any of these classes as the particles’ core is not composed of metallic magnetic elements but instead said elements are loaded, encapsulated or conjugated to the nanoparticle, such as Gd-conjugated dendrimers [181] or Mn-loaded liposomes [183,184].

In medicine, MNPs made up of a single metallic element are usually overlooked in favour of their oxide or alloy counterparts mainly due to the chemical instability these systems possess in vivo. They are highly reactive and easily suffer oxidation in the presence of water or oxygen meaning that in order to preserve their properties metal MNPs must be coated in a protective biocompatible layer [47]. However, this is not to say single metal MNPs do not possess their upsides. Despite the clear stability disadvantage, single metal MNPs offer a few advantages as well. For example, when compared to IONPs, simple Fe nanoparticles showed an ability to maintain their superparamagnetic properties at larger sizes and stability issues can be reduced with coatings [47].

However, these types of MNPs are far from the most studied as biomedical and diagnostic tools. One approach adopted to counter this limitation was the creation of nanoparticles containing multiple metallic elements, through a process named doping [18,47]. The different metals atoms chemically interact with each other and become more stable which improves their resistance to outside sources of chemical degradation [47]. Not only that, but the particle’s toxicity can also be mitigated through trading one metal in high quantities for multiple metals in lower quantities [21]. Moreover, the introduction of a second or third metal also changes the magnetic distribution of the atoms thus altering their magnetic behaviour with the goal of enhancing the MNP’s superparamagnetic properties without increasing the particle size [20,47]. For example, Pardo et al. synthetized IONPs simple-doped with Co, Mn, Zn or multi-doped with a Co-Mn and Co-Mn-Zn and observed that many of the MNPs exhibited better relaxation values than traditional Gd- and IONP-based contrast agents, suggesting a promising application as negative contrast agents [20].

Metal oxide MNPs can be composed of one oxide, such as IONPs, Gd_2_O_3_ nanoparticles or Mn_2_O_4_ nanoparticles, or by an oxide nanoparticle doped with a metal, with the latter focused on, although not limited to, metal-hybrid ferrite nanoparticles. Metal-hybrid ferrites are systems composed of the formula MFe_2_O_4_, where M is a transition metal that is associated with an Fe_x_O_y_ to form a nanoparticle. These hybrid MNPs bring an important advantage, an increase in particle saturation magnetization [21]. Saturation magnetization is the maximum magnetic moment per unit volume for a magnetic material, meaning it is a value in which increasing the applied external magnetic field will not increase the particle’s magnetization. Therefore, the higher its value, the easier it is to magnetize the MNPs. An additional advantage regards the presence of Fe. By replacing Fe with another element, less Fe is released into the bloodstream or other tissues [20].

However the inclusion of a second metal must not induce additional toxicity, otherwise it would defeat the purpose, as it sometimes happens with elements such as Co or Ni, where the particle’s cytotoxicity may even increase [20]. Therefore, a careful balance must be struck regarding the M/Fe_2_O_4_ ratio, in which M must also be carefully selected to assure safety of the final product in cytotoxicity assays [21]. Furthermore, as mentioned previously, the use of coatings that prevent ion leakage is a common effective tool in limiting toxicity [20]. Metal oxide nanoparticles, however, are not limited to metal-hybrid ferrites. Various nanoparticles made up of other metallic oxides, such as gadolinium oxides (Gd_x_O_y_), manganese oxides (Mn_x_O_y_) or cobalt oxides (Co_x_O_y_) have been developed and studied with the purpose of improving the efficacy and safety of previously existing MNPs [79,185,186].

### 5.1. Gadolinium

Gd is a member of the lanthanides and the most used metallic element in MRI. It possesses exceptional longitudinal water proton relaxation properties making it potentially the most suited element for T1-weighted MRI [79]. Currently, there are many contrast agent formulations in the market composed of Gd chelates, such as Magnevist^®^, Omniscan^®^ and MultiHance^®^. However, the search for alternative Gd formulations with higher sensitivity and a better safety profile is very much ongoing [76,187].

As an example, Gd_2_O_3_ nanoparticles are being studied as T1 contrast agents to both overcome the safety limitations concerning traditional Gd chelates and improve on the imaging properties of other MNPs, such as IONPs, which rely on the less preferable T2-weighted imaging [158,188]. Gd_2_O_3_ nanoparticles stay longer in the bloodstream and exhibit good biocompatibility. They can also be functionalized for active targeting, multimodal imaging, better biocompatibility and coupled with chemical drugs for treatment purposes. In various studies, these nanoparticles constantly showed T1-weighted MR signals comparable or even higher than Gd contrast agents currently on the market [188].

In addition, Gd (III) chelates have been successfully attached to dendrimers, especially polyamidoamine (PAMAM), with the goal of improving their relaxivity properties and blood circulation time [29,181]. Contrast agents with better relaxivity properties generate more discernible MRI images [189]. Other studies focused on entrapping Gd ions or chelates within nanocarriers, with fullerenes and carbon nanotubes being prime examples [29,181]. Fullerenes are usually icosahedral carbon cages (although they can take other forms) possessing a high surface area which can be used to, among other applications, encapsulate Gd compounds for imaging purposes. The Gd atom donates three electrons to the carbon structure bestowing paramagnetic properties onto the particle, which due to its high surface area produces a significant enhancement of particle relaxivity. Additionally, in pH ranging between 3 and 9, the fullerenes form aggregates further enhancing the T1 signal. Gd has been encapsulated in carbon nanotubes as well. These complexes tend to demonstrate better relaxivities than fullerenes and are also responsive to pH changes. Since the extracellular environment of cancer tissues tends to present lower pH values, this characteristic has inspired the synthesis of “smart” nanotubes for the detection of metastasized cancerous lesions [181].

Although more studies are needed, Gd nanotubes have also been studied in vitro for application on cell tracking of mesenchymal stem cells and macrophages [29,181]. It is also worth noting that many of the mentioned Gd nanocarriers exhibited better relaxivity values than Gd-based contrast agents currently on the market [181]. Safety of Gd-loaded carbon nanostructures is a topic that still requires more research, particularly regarding long-term administrations as carbon nanoparticles are known to accumulate in organs like the lungs, liver, spleen or kidneys for long periods of time (months or even years) leading to inflammatory and oxidative damage in these organs. Particle size may also influence clearance [29].

An interesting approach has been presented in the form of Gd upconverting nanoparticles. These are particles composed of rare metals that possess a unique property called photon upconversion. In photon upconversion, the nanoparticle absorbs two or more low energy photons sequentially, and afterwards, emits one photon of higher energy. In practice, absorption usually occurs within the infrared or near-infrared spectrum, and emission occurs in the visible or UV spectrum which can be measured and quantified by upconversion luminescence (UCL). Gd upconverting nanoparticles have excellent magnetic and optical properties and can easily be applied as multimodal imaging agents for T1-weighted MRI, UCL and CT. Preclinical studies suggest that these nanoparticles show great potential as imaging agents and they can act better than current Gd contrast agents [188].

### 5.2. Cobalt

Co is a transition metal with an essential role to play in human health and physiology as it constitutes a part of cobalamin, otherwise known as vitamin B12, an important cofactor in DNA synthesis and both amino acid and fatty acid metabolism. It is important for the normal functioning of the nervous system, myelin synthesis and maturation of red blood cells in the bone marrow. In low concentrations, patients can develop limb neuropathy and megaloblastic anaemia. Co has unique magnetic, optical, electrical and catalytic characteristics that make it suitable for a wide range of biomedical applications [190].

Two forms of Co_x_O_y_ are stable in nature, them being Co_3_O_4_ and CoO, with the former assuming the highest stability. Co ferrite (CoFe_2_O_4_) nanoparticles have been extensively studied on account of their magneto-crystalline anisotropy, high coercivity at room temperature and good saturation magnetization. They exhibit great physicochemical properties and good dispersibility. Unlike IONPs, which could lead to unwanted interactions with haemoglobin due to the release of Fe atoms, CoFe_2_O_4_ MNPs can help prevent leading to better penetration and hemocompatibility. They can also be doped with Zn, Mn or Ni to better improve the particle’s magnetic properties [191].

However, despite its superior magnetic characteristics, toxicity studies are being carried out in order to determine whether CoFe_2_O_4_ are a viable alternative to the classic IONPs and other MNPs since Co is more toxic than Fe [191]. Studies showed a reduced rate of cell proliferation and viability in areas of the body where CoFe_2_O_4_ nanoparticles were accumulated and suggested a cytotoxic effect that is dependent on nanoparticle concentration and cell type [191]. These nanoparticles have been shown to increase the production of ROS in many in vitro and in vivo toxicological studies, and, so far, studies have not established concrete evidence of the safety of CoFe_2_O_4_ nanoparticles [191,192,193,194]. Nevertheless, various coatings have been applied on their surface for the sake of improving particle stability and biocompatibility including citrate, mesoporous silica, alginate, poly(vinyl alcohol), poly(acrylic acid) and poly(ethanolimine), among others [191].

CoFe_2_O_4_ nanoparticles have many applications in medicine. Some in vivo studies investigate CoFe_2_O_4_ nanoparticles as contrast agents useful in both T1 and T2-weighted imaging [191]. Other studies investigate dual-mode imaging combining MRI with other techniques like photoacoustic imaging, particularly in the visualization of tumours. On a similar note, cell labelling has also been an area of interest. CoFe_2_O_4_ particles were used to track rat mesenchymal stem cells, macrophages and human gastric adenocarcinoma in model rats and mice. The particles were doped with various metals such as Zn, Mn and europium (Eu) and received coatings that enhanced target specificity or allowed for dual-mode T2-weighted MRI and fluorescence tracking demonstrating positive results as imaging agents [191].

Additionally, CoFe_2_O_4_ nanoparticles can be used for magnetic separation and isolation of biological substances in complex samples, such as cells and proteins. However, unlike contrast agents which are highly regulated based on the particle’s safety profile when inside the human body, these applications are not as bound by such restrictions and thus researchers are more able to focus efforts on a particle’s magnetic moment and ability to selectively bind to a specific target. Studies have shown considerable promise for CoFe_2_O_4_ agents in cell capture and protein isolation. Li et al. synthesized Cu^2+^ immobilized CoFe_2_O_4_ nanoparticles which could perform specific separation of bovine haemoglobin with a good adsorption capacity [195] and Sun et al. utilized CoFe_2_O_4_ nanoparticles contained in a microfluidic channel for immunomagnetic separation of mouse leukemic monocyte macrophage cells with a capture efficiency of 90% to 100% [196]. Furthermore, CoFe_2_O_4_ nanoparticles were also used to capture *Escherichia coli* and *Staphylococcus aureus* with an efficiency of 65% and 95%, respectively. However, it is worth noting that pathogen binding occurred in non-complex water samples as the purpose of this study was to evaluate CoFe_2_O_4_ nanoparticles as water treatment agents and not as compounds for magnetic separation in biological samples [197]. CoFe_2_O_4_ nanoparticles can also be incorporated into biosensors to increase sensitivity and specificity, create a less time-consuming procedure and lower the limit of detection. Biosensors are a crucial tool for diagnosis of several diseases through the measurement of specific biomarkers that can provide medical professionals with more complete information about the patient. For instance, the carcinoembryonic antigen (CEA) is a glycoprotein overexpressed in cancer and is commonly used to follow-up patients with colorectal cancer. Thus, an accurate measurement of this biomarker can result in an early diagnosis of cancer, which generally results in a better prognosis and less aggressive treatment options for the patient [191].

For this reason, Chen et al. has manufactured a high sensitivity immunosensor in containing CoFe_2_O_4_ nanoparticles conjugated with an anti-CEA antibody [198]. In a similar fashion, He et al. fabricated an immunosensor for detection of N-terminal pro-brain natriuretic peptide (NT-proBNP), an effective diagnostic and prognostic marker for heart failure exhibiting a wide detection range, high sensitivity and good reproducibility [199]. Furthermore, CoFe_2_O_4_ nanoparticles can also take part in biosensors for the detection of specific DNA strands to detect point mutations and single nucleotide polymorphisms without needing to turn to the time-consuming PCR methods [191].

### 5.3. Manganese

Aside from Gd, Mn has shown one of the highest magnetic properties. Mn is a cofactor for various enzymes such as Mn superoxide dismutase. It is vital for normal development, maintenance of nerve and immune cell functions and regulation of blood sugar and vitamins [200]. In order to create a safer alternative to Gd, Mn has been the subject of many studies resulting even in the clinical approval of two Mn-based contrast agents: mangafodipir and LumenHance™. Mangafodipir, commercial name Teslascan^®^, was approved for liver imaging and as an adjunct to MRI to aid in the investigation of focal pancreatic lesions. However, due to low sales and concerns over toxicity of Mn^2+^ ions, particularly in patients with liver failure, it was withdrawn from both USA and EU markets [186,201,202,203]. LumenHance™ is a MnCl_2_ loaded liposomal formulation approved as an oral contrast agent but was also withdrawn for similar reasons [184,204].

Mn_x_O_y_ nanoparticles have also been extensively studied, with focus on particles containing a core made up of MnO or Mn_3_O_4_, predominantly the former. With toxicity profiles and biocompatibility better than Gd-based agents, these inorganic nanoparticles are presented as excellent candidates for T1-weighted MRI, as well as fluorescent imaging, CT and theranostic applications [182,188]. Not only that but there have also been studies in which Mn nanoparticles demonstrated a superior T1 relaxation rate when compared to clinically available Gd-based contrast agents [186,205]. As is the problem with most nanoparticles, Mn nanoparticles accumulate in the liver and spleen after capture from macrophages of the MPS. Following degradation, Mn^2+^ ions are released in higher concentrations and can lead to toxic effects. However, many coating molecules can be conjugated with the nanoparticles to reduce capture by the MPS [182]. Conjugation with PEG has been particularly favoured given its ability to provide stealth and conjugate with specific polypeptides and aptamers to increase target specificity [182]. Other molecules provide Mn nanoparticles with better water solubility [182,186,206].

It is also possible to enhance the T1-T2 contrast of MNPs, by doping IONPs with Mn or formulating IONPs with the core surrounded in a Mn-containing shell [15]. However, since the T2 contrast Fe_x_O_y_ core generates a local magnetic field that opposes the spin alignment of the T1 contrast shell it can disturb the T1 relaxation process, causing the T1 signal to be diminished. Therefore, MNP doping appears to be a more viable strategy [15]. Another approach is the one adopted by Huang et al. who have synthesized pH-responsive dual mode IONPs coated by a Mn-based T1 contrast agent for effective cancer diagnosis and treatment. To avoid quenching the T1 signal the Mn contrast is released at the acidic environment of the cancer tissue to increase the distance between T1 and T2 contrasts while avoiding disturbance between them [81]. Furthermore, the morphology of the particle itself can highly influence its relaxation properties. An interesting approach has been the synthesis of octahedral nanoparticles instead of the classic spherical structure. Octagonal Mn nanoparticles possess a higher surface area which results in a significant enhancement of low-temperature ferromagnetic behaviour, thus enhancing their contrast ability. Usually, nanoparticles of a smaller size present better relaxation properties due to their superior superparamagnetic behaviour and higher concentrations of nanoparticles also improve image contrast, however, in a study by Douglas et al. the synthesized Mn octahedral nanoparticles presented similar T1 relaxation values to their spherical counterparts of a smaller size and higher concentration, which can point to a superior performance of octahedral contrast agents [207]. In order to create multimodal imaging particles, fluorescent dyes can be conjugated to the particle’s surface resulting in optical/magnetic resonance dual mode probes [182].

In terms of magnetic separation, Long and co-workers have worked on ways to effectively isolate and enrich phosphopeptides from complex samples. These are low-abundance peptides that may provide valuable information for early diagnosis of certain diseases. Mass spectrometry is often used to identify these peptides, however, because they are expressed at low concentrations detection is a difficult process. Therefore, MNPs have also been evaluated as potential tools to help boost phosphopetide signal and minimize impurity interferences [208,209]. In one study, synthesized CuFeMnO_4_ nanoparticles were effective in both neutral and acidic conditions [209]. In another study, Mn-doped Fe_2_O_4_ microspheres showed good reusability, dispersibility and selectivity towards phosphopeptides [208].

### 5.4. Dysprosium

Dysprosium (Dy), much like Gd, is part of the lanthanide series in the periodic table. It has one of the highest magnetic moments amongst all elements. For this reason, it is no surprise that Dy has been evaluated as a possible alternative to Gd given its ability to act as a T2 contrast agent. One of its biggest advantages is the capacity for ultra-high field MRI [182].

MRI scanner research is focusing more and more on generating higher magnetic field strengths (around 7 T or 9 T), which poses a problem since most currently used contrast agents are effective only at low magnetic field strengths (around 1.5 T or 3 T) [210].

The most common Dy nanoparticles for imaging purposes are Dy oxide and Dy fluoride, although Dy hydroxide have also been investigated. All of them are still in development stages [211]. DyF_3_ rhombus-shaped nanoparticles and NaDyF_4_ nanoparticles have been shown to possess remarkably high r_2_ (T2 relaxivity) values and show promise as effective negative contrast agents [6,212]. For multimodal imaging Dy_2_O_3_ nanoparticles can be doped with Terbium (Tb), another lanthanide, to create contrast agents with MR and optical imaging properties. Dy has also been used in Gd_2_O_3_ nanoparticle doping with the goal of promoting CT imaging, fluorescence and upgrading MRI capacity [6]. Furthermore, a study conducted by Veggle et al. demonstrated that the T2 relaxivity of NaDyF_4_ nanoparticles was dependent on particle size and magnetic field strength. An increase in particle size and magnetic field strength leads to an enhanced ability as a T2 contrast agent, however, T1 relaxivity showed little to no change [210].

### 5.5. Holmium

Another commonly investigated lanthanide that also displays great aptitude for ultra-high MRI, as well as a higher magnetic moment than Gd, is holmium (Ho) [213]. It has been used to perform doping on magnetic upconversion nanoparticles to create dual-mode MRI/OI (Optical Imaging) contrasts. Additionally, NaHoF_4_, HoF_3_ and Ho_2_O_3_ nanoparticles have been shown to be effective tools in negative contrast imaging [6,212]. Ni et al. reported r_2_ relaxation values of 222.6 mM^−1^s^−1^ at 7 T for NaHoF_4_ nanoparticles and Atabaev et al. produced PEGylated Ho_2_O_3_ nanoparticles presenting an r_2_ of 23.47 mM^−1^s^−1^ at 1.5 T, as well as green fluorescence due to intra 4f-transitions in Ho ions. Cytotoxicity studies were also conducted with the latter PEG-Ho_2_O_3_ nanoparticles which demonstrated nontoxicity at concentrations inferior to 16 μg/mL [213].

### 5.6. Other Lanthanides

Besides the already aforementioned elements (Gd, Dy and Ho), other lanthanides have been studied on their abilities as imaging agents, such as Eu, erbium (Er), Tb and ytterbium (Yb). Although not as common, some studies described the development of lanthanide oxide and lanthanide-doped nanosystems for MRI or multimodal imaging [214,215,216].

### 5.7. Silica

Silica nanoparticles on their own do not possess any special property that could be directly applied in bioimaging. However, they do exhibit a very tunable surface chemistry capable of binding to both organic and inorganic materials which can be used to integrate targeting molecules, magnetic chelates (on MRI), fluorescent dyes (on OI) and radioisotope tracers (on PET) for development of single and multimodal imaging agents. Silica nanoparticles can be divided into two major groups: mesoporous silica nanoparticles (MSNPs) and solid silica nanoparticles (SSNPs), both being attractive options for bioimaging agents. MSNPs, such as MCM-41 or SBA-15, are made up of a honeycomb-like porous matrix of empty pores and channels in which various molecules can be loaded. In the past decade, MSNPs have attracted considerable attention by virtue of their inherently large surface area and pore volume useful for functionalization and controllable particle size [217].

MSNPs have been combined with magnetic materials such as Gd chelates and Mn [218] for MRI purposes. In a study conducted by Lin et al. a Gd chelate was loaded into MSNPs obtaining superior T1 and T2 relaxivity values than SSNPs coated with a similar Gd contrast agent. The nanoparticle also was successfully applied as a T1 contrast agent for visualization of the aorta in a DBA/1J mouse as well as a T2 contrast agent for liver imaging. In a similar fashion, Chen et al. synthesized hollow MSNPs decorated with Mn_x_O_y_ nanoparticles as a multimodal pH-responsive T1-weighted MRI and ultrasonography contrast agent. Within the tumour acidic environment, the loaded Mn nanoparticles start dissolving and releasing Mn^2+^ ions which in turn produced a localized T1 enhancement in the tumour site [219].

On a related note, SSNPs (which are non-porous particles) have also been investigated as MRI contrast agents. Kobayashi et al. produced a colloid solution of Gd-based positive contrast agents immobilized in spherical silica particles. The resulting relaxivity value was comparable to that of the commercial Gd-based contrast agent Magnevist^®^ [220].

The fact that silica can bind to so many molecules is also explored in the area of magnetic separation. Many studies cover the use of IONPs covered in a silica shell which is then functionalized with various coatings that increase the particle’s ability to bind to specific molecules [106,221]. For example, Bai et al. and Kang et al. developed amino-functionalized MNPs for DNA separation from complex samples. In the latter, the MNPs showed a DNA adsorption efficiency 4 to 5 times higher when compared to simple silica-coated IONPs [106,221]. Other studies focused on protein separation [222,223]. Another interesting study focused on SiO_2_ nanoparticles coated in various layers of IONPs and silica. The rationale being that IONPs show very slow accumulation and low separation yield when an external magnetic field is applied. Therefore, various layers of IONPs conjugated to the larger SiO_2_ nanoparticles is thought to accelerate particle accumulation, creating a quicker, more effective separation process. The silica coatings also grant stability and promote water solubility [224].

### 5.8. Copper

Copper (Cu) nanoparticles have emerged as potential theranostic agents for cancer management in the form of copper sulfide nanoparticles (CuSNPs). The diagnostic imaging properties of CuSNPs originate, in great part, due to their ability to absorb electromagnetic waves in the Near-infrared (NIR) spectrum which is useful for Photoacoustic Imaging (PAI). However, the magnetic properties of CuSNPs seem to be lacking as most studies focusing on MRI included a metal chelate conjugated to the particles. For instance, Liu et al. developed CuSNPs functionalized with PEG and conjugated to Mn (II) chelates for in vivo tracking and quantification of human breast cells in tumour bearing mice [225]. Zhang et al. synthesized CuSNPs functionalized with thiol-PEG and the Gd-based chelate Gd-diethylenetriamine pentaacetic acid (DTPA), which resulted in particles exhibiting a T1 relaxivity coefficient value two times higher than conventional DTPA [226,227]. Cu also has applications in the magnetic separation of haemoglobin, as it promotes hydrophobic and metal-affinity interactions with the histidine residues of proteins, especially haemoglobin. For this reason, researchers have synthesized IONPs with a Cu-containing shell for the purpose of purifying and enriching haemoglobin from blood samples [228,229].

### 5.9. Metal Alloy MNPs

Metal alloy MNPs are synthesized by combining two or more different pure metallic elements into the particle’s core. The surface modification of many alloy nanoparticles has improved their solubility by allowing the binding with carboxylate- or amine-based surfactants, such is the case of FePt nanoparticles [47]. The greatest advantages behind alloy MNPs in the diagnostic field are the improvement of their magnetic properties, ability to include other properties useful for multimodal imaging and additional protection against chemical degradation in vivo (e.g., oxidation) [47,230,231,232]. For example, nanoparticles with an FeCo core can be viewed as an approach for maximizing the saturation magnetization of MNPs. Song and colleagues developed FeCo nanoparticles coated with graphitic carbon which protects the core metals from chemical degradation in aqueous environments. In addition, this system was later superficially modified with PEG to increase half-life [233].

The particles presented a six-fold higher relaxivity signal than VivoTrax^®^ (Alameda, CA, USA), a commercialized tracer, and a ten-fold higher signal than Feraheme^®^ as well as a high optical absorbance in the PAI resulting in a promising dual-mode MRI and PAI agent [233]. Torresan et al. developed FeAu nanoparticles coated with thiolated PEG to be used as a CT/MRI contrast agent in mice. Safety in vivo studies showed low particle accumulation in the liver and spleen and a better clearance profile than simple Au nanoparticles, with no evidence of toxic effects. Efficacy wise, MRI yielded results similar to those of the commercially approved Endorem^®^ and CT measurements proved better than the clinically available contrast agent, iopromide [231]. Other alloy MNPs such as FeNi and FePt have also been evaluated as MRI contrast agents demonstrating high superparamagnetic properties and low toxicities [234]. Alloy MNPs possess high magnetophoretic mobility and, as such, are being assessed as possible replacements for IONPs in magnetic separation. In a study conducted by Hutten et al. FeCo MNPs presented the highest magnetophoretic mobility values, even when compared to IONPs [232]. Nevertheless, more research is needed on the efficacy, and especially safety, of alloy MNPs in bioimaging and magnetic separation.

## 6. Conclusions

MNPs are inherently endowed with features that can be applied in disease diagnosis, particularly as imaging contrast agents and allowing magnetic isolation of biomolecules for diagnostic assays.

MRI is a widely used non-invasive diagnostic technique with an excellent safety, however, it still lacks some sensitivity, specificity, and possesses some safety issues regarding exposure of certain risk groups to Gd-based contrast agents. To overcome these limitations, research has focused on a number of options, with MNPs at the forefront of all these different approaches namely: (a) formulating better contrast agents that will aid in creating more discernible MR images; (b) investigating the use of multimodal imaging techniques, such as MRI-PET, MRI-OI or MRI-CT to achieve a more complete image combining the advantages of both imaging methods while reducing their limitations; (c) application of ultra-high magnetic fields which could create discernible images with less concentration of contrast agents; (d) testing alternatives to the classic Gd chelates commonly used in clinical practice.

MNPs can also be applied in separating and isolating specific cells and biomolecules from a complex sample. Currently used protocols which do not employ MNPs are time-consuming and purification steps require expensive and sensitive equipment. MNPs shorten the purification stage and eliminate pre-treatment and pre-enrichment steps.

Overall, MNPs are expected to aid in developing improved analysis protocols that are faster, cheaper and simpler than currently existing ones.

## Figures and Tables

**Figure 1 nanomaterials-11-03432-f001:**
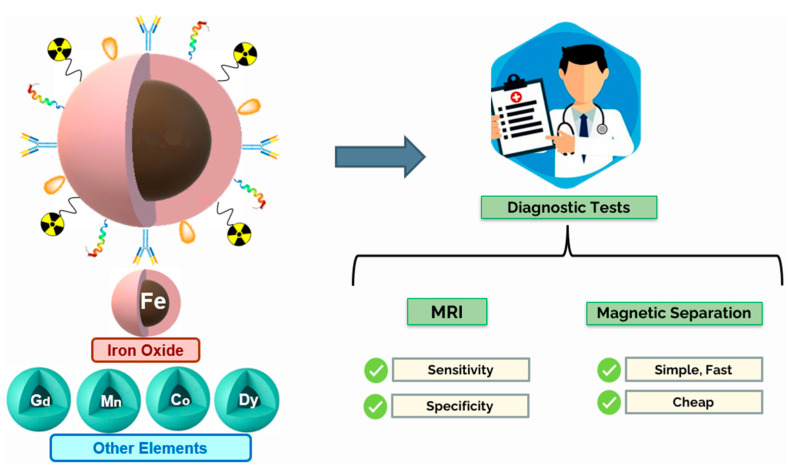
Magnetic nanoparticles as a tool to develop effective, sensitive, specific, simple and fast diagnostic tests to be applied in different imaging techniques.

**Figure 2 nanomaterials-11-03432-f002:**
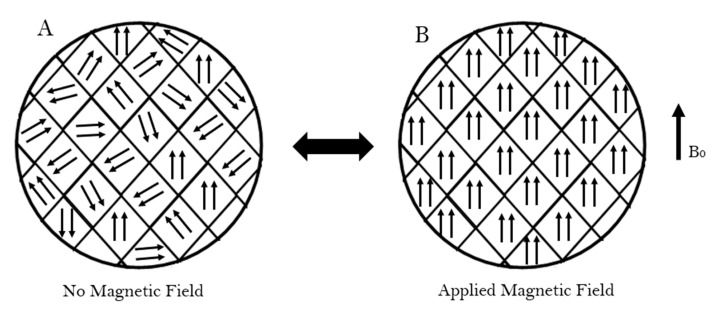
Schematic representation of the various magnetic domains that compose the MNP crystalline structure in the presence and absence of an external magnetic field.

**Figure 3 nanomaterials-11-03432-f003:**
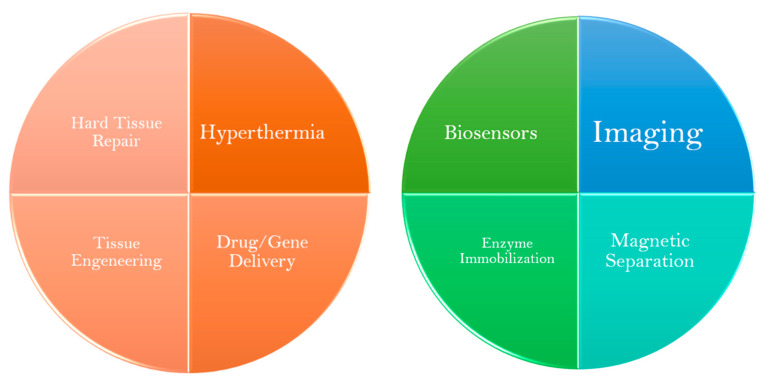
Different biomedical applications of MNPs, including therapeutic applications (**left**) and diagnostic applications (**right**).

**Figure 4 nanomaterials-11-03432-f004:**
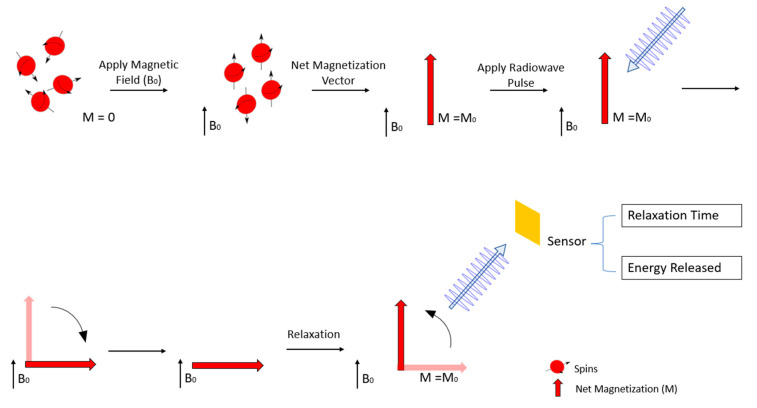
Schematic representation of the physical process occurring during an MRI examination.

**Figure 5 nanomaterials-11-03432-f005:**
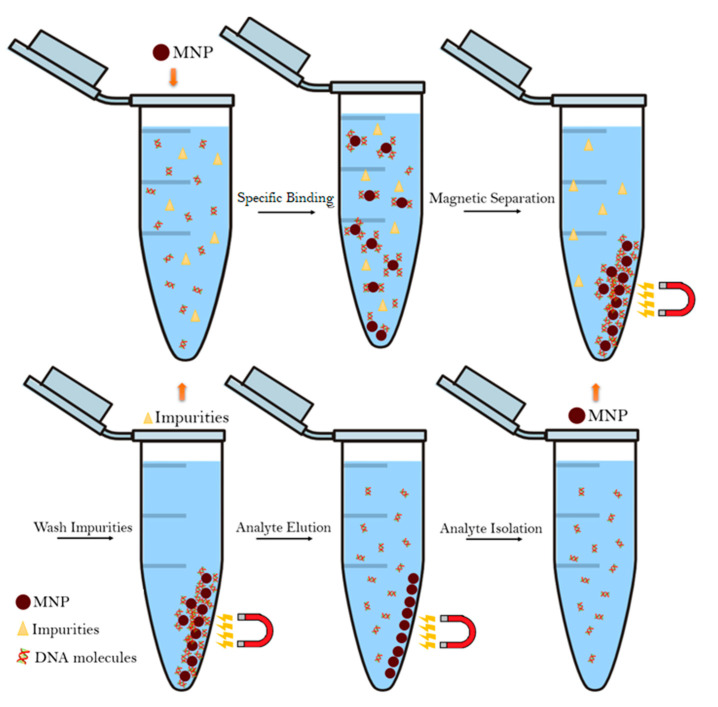
Schematic representation of the magnetophoretic phenomenon used in the magnetic separation of biomolecules.

**Figure 6 nanomaterials-11-03432-f006:**
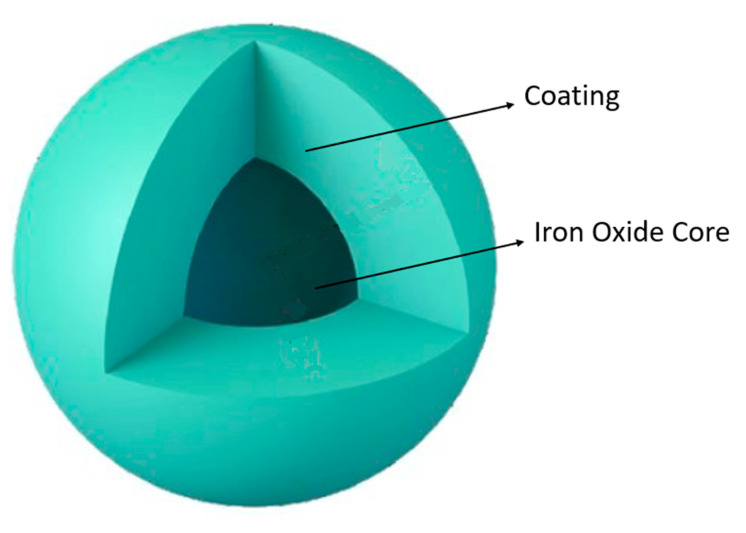
Three-dimensional model of an IONP. Adapted from [122] under the Creative Commons Attribution License: https://creativecommons.org/licenses/by/4.0/.

**Figure 7 nanomaterials-11-03432-f007:**
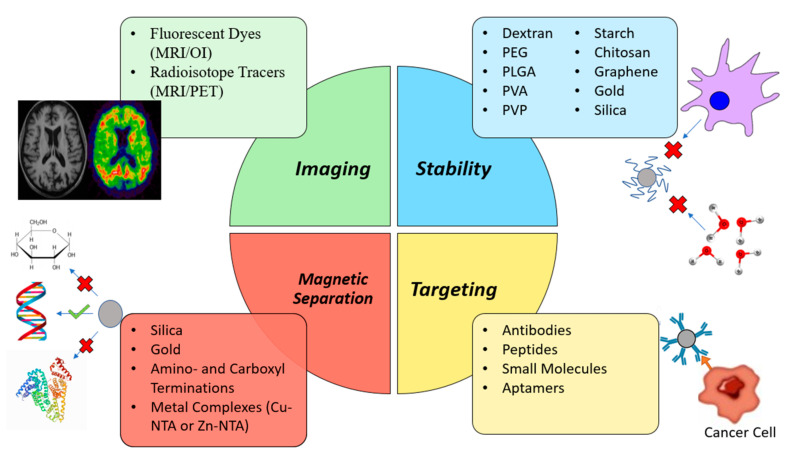
Schematic representation of some of the coatings used in MNPs and their specific applications.

**Figure 8 nanomaterials-11-03432-f008:**
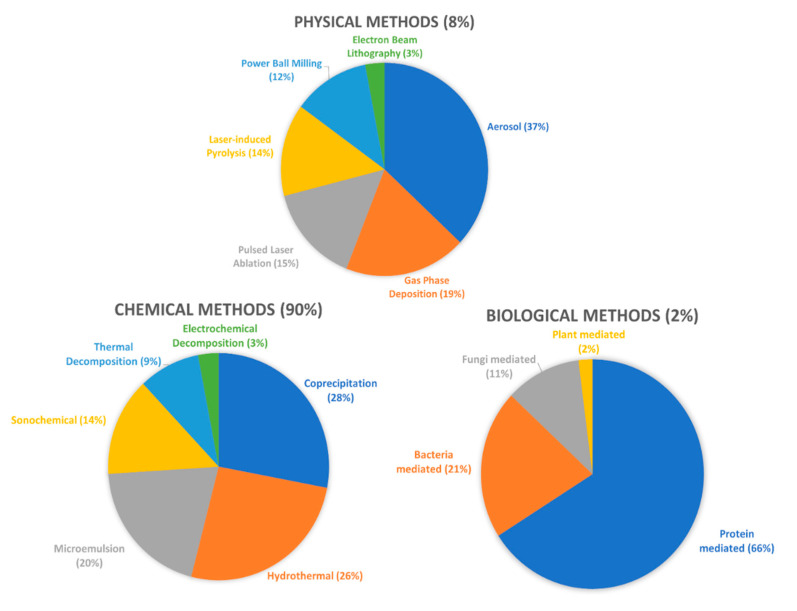
Synthesis of MNPs using physical, chemistry and biological methodologies. Adapted and reproduced with permission from Ali et al. [26].

## Data Availability

The data presented in this study are available on request from the corresponding author.

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
