# Peer review of "A Comprehensive Updated Review on Magnetic Nanoparticles in Diagnostics"

_nanomaterials, 2021, doi:10.3390/nano11123432_

Round 1
Reviewer 1 Report
Proposed review constitutes an interesting work presenting numerous information concerning mainly magnetic nanoparticles. Nonetheless, some elements needs to be improved to increase the scientific value of the paper – all suggestions are given in more detail below:
- Section Introduction: this part presents interesting information but it is suggested to change the order of the first and second paragraphs. Firstly, brief information concerning magnetic nanoparticles should be given and then the explanation why exactly these particles constitutes an interesting solution/alternative for diagnostic purposes.
- Paper should be written in a third person, not in a first one (i.e. instead of phrases “we present”, a phrase “sth has been presented” (or “was presented”) should be used.
- Each abbreviation should be explained when it is used for the first time, e.g. FDA or EMA from section 2 are explained only in section 4.3.
- Section 3.1.: first sentence sounds “From a general point of view and in medical field, nanoparticles are colloidal systems 101 sized between 1 and 1000 nm “. Please, check carefully this information because in general size “nano” refers to materials which at least one dimension is within the range 1-100 nm, not 1000 nm. The same applies to the information that “Usually, MNPs should have a mean size below 100 nm in order to exhibit superparamagnetic properties”. These properties are shown by smaller particles.
- Line 139: magnetocaloric effect showed by magnetic nanoparticles needs to be explained in more detail.
- Line 146: the notation such as “Fe, Co and nickel (Ni)” should be unified. Authors should use only the symbols of the elements or in each case the name of the element and its symbol in brackets.
- The notation of polymer names should be corrected, e.g. “poly(ethylene glycol)” should be written instead of “polyethylene glycol” (line 184).
- It is strongly suggested to supplement the review with additional subsections in which the toxicity of magnetic nanoparticles and their degradation will be briefly discussed.
- Paper contains some language and grammar mistakes which should be corrected (e.g. “actively delivering drugs” instead of “active delivering drugs” – line 193).
- Line 273 – it should be “it is” instead of “itis”.
- Section 3.2.2.1.: last paragraph concerning the separation of nucleic acids needs to be discussed more widely.
- Final conclusions should be more brief.
Author Response
The authors would like to express our profound gratitude to reviewer for the constructive suggestions made to our paper.
We carefully considered all comments and questions made and completed a thorough point-by-point response (PDF Version - Reviewer 1 responses). All revised sections are highlighted in green.

Reviewer 2 Report
The manuscript" A comprehensive updated review on magnetic nanoparticles in diagnostics" by Gaspar group described the MNP with various other metallic elements (Co, Mn et al.) in diagnostics. While this work is thoroughly done, I believe the paper will be of interest to the readership of nanomaterials and would recommend it for acceptance after the minor points.
Comments:
- The protein purification and enzyme immobilization and cell
separation using MNPs was unrelated diagnosis. Suggest to remove
3.2.2.2 and 3.2.2.3 sections to match the paper title.
- line 988, MR should be MRI.
- In conclusion section, please shorten the text.
Author Response
Author’s response to Reviewer 2:
The authors would like to express our profound gratitude to Reviewer 2 for the positive feedback. We carefully considered all minor points.The revised sections are highlighted in green in the revised manuscript.
Comments:
- The protein purification and enzyme immobilization and cell separation using MNPs was unrelated diagnosis. Suggest to remove 3.2.2.2 and 3.2.2.3 sections to match the paper title.
Reply: The sections 3.2.2.1., 3.2.2.2. and 3.2.2.3. include the methodologies for the preparation of samples for diagnosis using MNPs.
In this sense we think they match the paper title. Nevertheless, in section 3.2.2.2 the information regarding enzyme immobilization was removed and this title was changed accordingly.
- line 988, MR should be MRI.
Reply: This term was corrected.
- In conclusion section, please shorten the text.
Reply: The conclusions were drastically reduced.
Reviewer 3 Report
The review by Gaspar and colleagues provides a very detailed overview of applications of magnetic nanoparticles (MNP) for diagnostics. The review provides a lot of detailed information that should be of interest to scientists working in this area or considering to do so. Publication is recommended after consideration of the questions and suggestions below.
- Introduction, Figure 1. It is not clear to me how MRI, even with improved contrast agents, can be considered “cheap”.
- Section 3.1. The range of 1 to 1000 nm seems too large for nanoparticles in the medical range, even though it is recognized that the 100 nm limit is too restrictive for medical applications.
- There are two line breaks in the paragraph starting on line 135, which do not appear to be new paragraphs. This occurs in multiple locations throughout the review and should be corrected.
- Line 167. Which metal-based nanoparticles have reached clinical trials?
- Section 3.2.1, line 326. Approximately how many MNP are currently commercially available as contrast agents?
- Figure 8. The molecule (should be molecular) recognition section is somewhat confusing. The explanation mixes surfaces that can be easily modified (gold, silica) with methods for introducing functionality that provides molecular recognitions elements.
- Figure 9. The text (lines 544-5) indicates that coprecipitation is the most widely utilized method. However, the percentage of hydrothermal methods is very similar.
- Section 4.3. It is not clear in the initial paragraphs if all the IONP formulations are approved for imaging applications. It would be useful to refer to SI Table 3 much earlier in this section.
No information is presented on the type of iron oxide used in the various formulations. Please consider whether this should be added.
- Lines 587-89 are unclear. What is the relationship between Ferucarbotran and Supravist? Similarly the description of Sienna (590-596) is confusing. The IONP is not a device, presumably it is the combination of Sienna with Sentimag that is the device?
- The paragraph starting at line 600 notes examples of IONPs that are approved for oral administration. The delivery mechanism for the materials discussed earlier in this section is not specified; an administration method should be added for clarity.
- Section 4.4 provides a lot of detail on IONPs, but then section 4.5 basically says no one uses them. This seems a bit odd and the authors should consider whether some modification of either section is required. These 2 sections suggest that IONPs are not really used for biosensing applications. Is that the case?
- Section 5.1 notes that gadolinium is the most commonly used metal for MRI. It would be useful to note here the different types of gadolinium species that are currently used (are they all chelates?). In fact it might be even clearer to mention this earlier in the review as a motivation to study a variety of nanoparticle contrast agents. This section also mixes nanoparticles with chelates to some extent which is somewhat confusing to the non-expert.
- Section 5.2 on cobalt focuses entirely on Co ferrite, after a brief introduction to the function of cobalt in human health. Perhaps this section should have a different title? Also this is the first example where there is much discussion of separation/sensing application of MNPs. Perhaps some perspective on the types of materials used for separation and sensing should be noted in Section 3.
- There is a statement on line 905 that MnxOy NPs have better toxicity profiles than Gd-based agents. This seems somewhat at odds with the statement in lines 898-899 on the withdrawal of a Mn based reagent. The section on Mn is confusing on the utility of Mn-based reagents and their comparison to Gd.
- Section 5.4. Please define r2. Also be clear on whether any Dy reagents are approved or they are still in the development/research stage.
- Section 5.4, 5.5 and 5.6 are all short and it might be better to combine all 3 under the heading of “Other lanthanides”.
- The paragraph starting on line 1137 of the Conclusions is not clear on whether any MNPs are really a viable alternative for cell separation, despite the final statement that they are in clinical use. Please clarify.
- Typos.
Abstract, line16: challenges that cause difficulty with their entry….
Line 371. Are gathered towards
Line 1072: have also been evaluated

Author Response
Comments and Suggestions for Authors
The review by Gaspar and colleagues provides a very detailed overview of applications of magnetic nanoparticles (MNP) for diagnostics. The review provides a lot of detailed information that should be of interest to scientists working in this area or considering to do so. Publication is recommended after consideration of the questions and suggestions below.
Author’s response to Reviewer 3:
The authors would like to express our profound gratitude to Reviewer 3 for the constructive suggestions made to our paper.
We carefully considered all comments and questions made and completed a thorough point-by-point response (PDF version reviewer 3 responses).

Reviewer 4 Report
The manuscript: A comprehensive updated review on magnetic nanoparticles in diagnostics extensively presents an overview of the magnetic nanoparticles used for diagnostic in biomedicine. The work describes not only the main characteristics of this kind of nanoparticles, but also their applications in medicine. I have really appreciated the structure of the work in two parts. The authors firstly present the research and information gathering methodologies adopted as well as the criteria used to determine which publications presented the information with the most interest. In a second part, the authors describe the main properties of magnetic nanoparticles as well as their applications in biomedicine, concretely in diagnosis.
The manuscript perfectly describes the state of the art of the interesting field constituted by the use of magnetic nanoparticles in diagnosis. It also offers a wide and completed overview of the different types of magnetic nanoparticles, their compositions and their properties. Finally, the authors analyze the use of this magnetic nanoparticles in different medical techniques for diagnosis.
I recommend the publications of this review. The only suggestion that I propose is to include a more exhaustive analyses of the nanoparticle functionalization process. As it is well known, nanoparticles used for biomedical applications have to achieve some fundamental requisites: low cytotoxicity, colloidal stability, and the possibility to be subsequently bioconjugated with other biomolecules. An optimal design of nanoparticles suitable for biomedical applications requires a proper functionalization, being a key step in the synthesis of such nanoparticles, not only for subsequent crosslinking to biological targets, but also to endow these materials with colloidal stability and to avoid cytotoxicity. In this sense, a reliable characterization of the functionalization process effectiveness would be therefore, crucial for subsequent bioconjugations. It is also well known that this process constitutes one of the most difficult steps of nanoparticles fabrication process. Although this work is focused on the diagnosis applications and not on nanoparticles preparation, the success of process frequently depends on the proper functionalization of nanoparticles. Consequently, I would include a small section describing the importance of this aspect and analyzing the alternatives. In connection with this, the term “shell” is not always clear in the text (for example in figure 7) This term is commonly kept for nanoparticles with a structure formed by two different compositions or phase in the core of the nanoparticle. The authors sometimes use this term for the capping used to functionalize the nanoparticle core.

Author Response
Reviewer 4
Comments and Suggestions for Authors
The manuscript: A comprehensive updated review on magnetic nanoparticles in diagnostics extensively presents an overview of the magnetic nanoparticles used for diagnostic in biomedicine. The work describes not only the main characteristics of this kind of nanoparticles, but also their applications in medicine. I have really appreciated the structure of the work in two parts. The authors firstly present the research and information gathering methodologies adopted as well as the criteria used to determine which publications presented the information with the most interest. In a second part, the authors describe the main properties of magnetic nanoparticles as well as their applications in biomedicine, concretely in diagnosis.
The manuscript perfectly describes the state of the art of the interesting field constituted by the use of magnetic nanoparticles in diagnosis. It also offers a wide and completed overview of the different types of magnetic nanoparticles, their compositions and their properties. Finally, the authors analyze the use of this magnetic nanoparticles in different medical techniques for diagnosis.
I recommend the publications of this review.
The authors would like to express our profound gratitude to reviewer for the positive feedback.
We carefully considered all comments and suggestion made. Thank you.
The only suggestion that I propose is to include a more exhaustive analyses of the nanoparticle functionalization process. As it is well known, nanoparticles used for biomedical applications have to achieve some fundamental requisites: low cytotoxicity, colloidal stability, and the possibility to be subsequently bioconjugated with other biomolecules. An optimal design of nanoparticles suitable for biomedical applications requires a proper functionalization, being a key step in the synthesis of such nanoparticles, not only for subsequent crosslinking to biological targets, but also to endow these materials with colloidal stability and to avoid cytotoxicity. In this sense, a reliable characterization of the functionalization process effectiveness would be therefore, crucial for subsequent bioconjugations. It is also well known that this process constitutes one of the most difficult steps of nanoparticles fabrication process. Although this work is focused on the diagnosis applications and not on nanoparticles preparation, the success of process frequently depends on the proper functionalization of nanoparticles. Consequently, I would include a small section describing the importance of this aspect and analyzing the alternatives. In connection with this, the term “shell” is not always clear in the text (for example in figure 7) This term is commonly kept for nanoparticles with a structure formed by two different compositions or phase in the core of the nanoparticle. The authors sometimes use this term for the capping used to functionalize the nanoparticle core.
Reply: We understand the reviewer’s suggestion. We have presented a brief information on the various coatings and their specific applications in the manuscript (section 4.1.). As the reviewer has stated, our work focuses more on the applications of these coatings (especially in diagnostics) and not the processes used to coat the nanoparticles.
Regarding the term “shell”, The reviewer makes a compelling argument. The terms “coating” or “surface layer” or “biocompatible layer” are now used for the capping used to functionalize the nanoparticle core. The term “shell” is now used for two different phases in the core.